# A dynamic Bayesian network model for predicting organ failure associations without predefining outcomes

**Roberto Alberto De Blasi**[1]*, **Giuseppe Campagna**[1], **Stefano Finazzi**[2]

**1** Dipartimento di Scienze Medico-Chirurgiche e Medicina Traslazionale, Università degli studi di Roma Sapienza, Ospedale Sant'Adrea, Rome, Italy, **2** Laboratorio di Clinical Data Science, Dipartimento di Salute Pubblica, Istituto di Ricerche farmacologiche Mario Negri IRCCS, Ranica (BG), Italy

* radbl@libero.it

**Data Availability Statement:** The datasets generated and/or analyzed during the current study are anonymously available on the following

## Abstract

Critical care medicine has been a field for Bayesian networks (BNs) application for investigating relationships among failing organs. Criticisms have been raised on using mortality as the only outcome to determine the treatment efficacy. We aimed to develop a dynamic BN model for detecting interrelationships among failing organs and their progression, not predefining outcomes and omitting hierarchization of organ interactions. We collected data from 850 critically ill patients from the national database used in many intensive care units. We considered as nodes the organ failure assessed by a score as recorded daily. We tested several possible DBNs and used the best bootstrapping results for calculating the strength of arcs and directions. The network structure was learned using a hill climbing method. The parameters of the local distributions were fitted with a maximum of the likelihood algorithm. The network that best satisfied the accuracy requirements included 15 nodes, corresponding to 5 variables measured at three times: ICU admission, second and seventh day of ICU stay. From our findings some organ associations had probabilities higher than 50% to arise at ICU admittance or in the following days persisting over time. Our study provided a network model predicting organ failure associations and their evolution over time. This approach has the potential advantage of detecting and comparing the effects of treatments on organ function.

## Introduction

Since the early 2000s, Bayesian networks (BNs) have attracted considerable interest in the field of medicine [1] for their ability to model complex systems by learning the network structure among variables from observed data, thus providing an interpretation of causal relationships among variables instead of merely capturing associations [2]. In critical care medicine, BNs have been applied clinically to investigate the complex relationship among failing organs. Since the 1990s, many criticisms have been raised regarding the use of mortality to reflect the efficacy of a treatment, particularly when the disorder under consideration has limited lethality

repository: DANS.https://doi.org/10.17026/dans-zgz-7keg. DATASET SOFA.

**Funding:** Giuseppe Campagna received a fund for data analysis from the research funds of the Dipartimento di Scienze Medico-Chirurgiche e Medicina Traslazionale of University of Rome "La Sapienza" for the analysis of data (n. 000323_19 master del.04.03.020; PI: GC). The funders had no role in study design, data collection and analysis, decision to publish, or preparation of the manuscript.

**Competing interests:** The authors have declared that no competing interests exist.

[3]. Despite these concerns, in most clinical trials resorting to predefined outcomes continues to be the predominant criterion for evaluating whether a therapeutic action is effective [4].

In this work, we considered the onset of organ failure and used BNs to determine the probabilities of connections among failing organs that could be assumed to reflect the effects of therapeutic actions.

The use of dynamic Bayesian networks (DBNs) added to BNs is beneficial for investigating the temporal order and duration of organ failures, helping to predict the most likely progression during a patient's stay in the intensive care unit (ICU) [5, 6].

To make DBN prediction reflective of the pathophysiological process of organ failure in critically ill patients without constraining the network structure with *a priori* assumptions, we adopted an approach that differed from the procedures commonly employed for BN applications in health care. This process normally entails a two stage process to assess conditional probabilities [7]. The first stage is the identification of possible dependence relationships between variables. This stage involves manually defining causal relationships represented by directed arcs between network nodes. The second stage includes the identification of qualitative probabilistic and logical constraints, reducing the number of parameters to be estimated. On the one hand these procedures have the advantage of making the network robust and clinically interpretable on the basis of existing knowledge. On the other hand, imposing ordering and constraints on the probabilistic relationships between organs and processes can limit the ability of the model to reflect the data.

Given that multiple organ dysfunction syndrome (MODS) arises from a widespread septic or non-septic inflammatory reaction involving tissue microcirculation the associations and sequences of organ failure events are not shaped by trivial causal relationships or constraints [8, 9].

Contrary to previous studies that used DBNs to predict the dynamics of failing organs by hierarchizing organ interactions and forcing discrete outcomes [6, 10], our study aimed to develop a model for identifying interrelationships among organs without defining a specific outcome as a reference and without hierarchizing organ interactions. Since relationships among failing organs are complex and not completely known, we decided to learn the causal structure of a short-term DBN from data with Markovian constraints enforced through a blacklist of temporal relationships. In addition, we built a DBN by learning the associated structure and estimating parameters connected with conditional probabilities. We initially included an extensive set of organs and data points and gradually reduced them to a minimum clinically relevant group.

## Materials and methods

We included data retrospectively collected from the "Prosafe" database of the Italian Group for the Evaluation of Interventions in the Intensive Care Units (https://givitiweb.marionegri.it). Data collection was approved by the the ethics committee of the Sapienza University of Rome at the Sant'Andrea University Hospital (Ref. 3408 2014/09.10.2014, Prot. 1244/2014), and all subjects provided written informed consent. Data were derived from patients admitted to the adult intensive care units (ICUs) of the Sant'Andrea University Hospital in Rome, the A. Manzoni Hospital in Lecco and the Di Cristina-Benfratelli Civic Hospital in Palermo, Italy, from January to September 2013. We collected data from patients with at least two organ failure events and a hospital stay longer than 48 hours so that the organ interactions and their progression could be assessed.

For patients who had multiple ICU admissions, we considered only the first admission. During their ICU stay, patients were treated according to the usual clinical practice of the

period of data collection and received organ function support when needed (i.e., mechanical ventilation, hemodialysis and/or vasopressors).

For the DBN construction, we included as nodes the same organs considered for the Sequential Organ Failure Assessment (SOFA) score [11], and the observation times ($t$) were the days of data collection for the included organs and systems: cardiovascular$_t$ (CV), respiratory$_t$ (lung), central nervous system (CNS)$_t$, renal$_t$ (kidney), liver$_t$ (Liver) and coagulation$_t$ (C). The SOFA score was computed using data collected from laboratory tests, cardiovascular monitoring, vasoactive drug dosages and clinical reports. A failure event for an organ was defined as a SOFA score greater than or equal to two [12]. For each patient, we collected daily data recordings for 7 consecutive days after ICU admission through an *ad hoc* electronic case report form. This interval was chosen because it approximatively matches the patients' mean ICUs stay, as previously reported (https://givitiweb.marionegri.it). A day was defined as a 24 h period starting at 12.00 a.m. except for the first day (*t0*), the day of ICU admission. If a patient was admitted before 12:00 p.m., *t0* was defined as the 24 h period already in progress, starting at 12:00 a.m.; if the time of admission was after 12:00 p.m., the remaining hours until 12:00 a. m. were pooled with the following day.

## Bayesian networks

A Bayesian network is a probabilistic directed acyclic graph depicted as nodes, which represent random variables, and arcs between nodes, which express the probabilistic dependencies between variables. The direction of the arc (arrow) between two nodes, A and B, establishes a "parent" node (A) and a "child" node(B).

DBNs extend BNs by encoding the temporal or spatial evolution of variables expressed by repeated time series models [13, 14]. Assuming *n* random variables, $X = X^1, X^2, \ldots, X^n$, we constructed a DBN by adding a node (*i*) for each variable at each time step $t$ ($X_t^i$). For a dynamic model, we assumed that the state of the system at $t$ would affect the future state of the variables at $t+1$ and would depend on the previous configuration at $t$-1 [15]. Causal relationships follow the arrow of time: the state of any node at a given instant can influence the states of nodes in the future but never in the past [16]. Furthermore, we assumed that the Markov property held true, implying that the stochastic process underlying the onset of organ failure was memoryless. As a consequence of this assumption, states of nodes in the network at time $t$ depend exclusively on nodes states at $t$ and $t-1$ (but not at $t-j$, with $j > 1$) [17].

## Dynamic Bayesian network formulation

All computations were performed using R software and the package bnlearn (https://www. bnlearn.com/releases/bnlearn_latest.tar.gz). The network structure was learned from data using the hill-climbing algorithm [18] starting from an empty network (a network with no arcs). We used a blacklist to exclude unreasonable temporal causal relationships (arcs directed from the future to the past) and enforce the Markovian properties (arcs could link nodes only at the same time or between time $t$ and $t-1$). The score used in the optimization process was the Bayesian information criterion.

To obtain a more robust model with higher predictive performance [19], we constructed an averaged network through bootstrapping. We generated 500 realizations of the network structure through the function bn.boot with the hill climbing algorithm [20]. Arc strength was computed as the fraction of realizations in which the arc was selected by the hill-climbing algorithm. Arc direction was the fraction of realizations in which the arcs had a given direction. We retained only arcs with directions greater than or equal to 0.5 and with strengths greater

than an optimal threshold, automatically determined based on the shape of the cumulative distribution of arc strengths [19].

Once the network structure was determined, the probability matrices representing the probability of each node, conditional on the value of its parent node were estimated with a maximum likelihood algorithm using the function bn.fit. We considered 4 possible DBNs with different number of nodes.

In the first network, we used variables for all organs (6 nodes) for eight days. Given that the platelet count changes according to the duration and severity of sepsis and mostly cannot be affected by a specific treatment, node "C" was omitted from the second network. We also tested a third network including 6 nodes at three time points: $t0$ = day of ICU admission, $t2$ = $2^{nd}$ day and $t7$ = $7^{th}$ day. This network was also tested without the platelet count (C node). To test the model reliability, we applied a $k$-fold cross-validation analysis to this final network, with $k$ = 10. We computed the sensitivities and specificities of the child node predictions from parent nodes.

## Results and discussion

### Characteristics of patients and networks

We enrolled 850 patients ranging in age from 18.0 to 95.0 years (median: 69.0), with a mean of 65.7 ± 15.9. Of these, 536 (63.1%) were male (mean age 65.6 ± 15.5 years, range 18.0–95.0), and 314 (36.9%) were female (mean 65.9 ± 16.7 years, range 18.0–93.0). The number of discharged patients was 651 (76.6%), of whom 410 were male (63.0%) and 241 were female (37.0%). The mean age of the males did not differ significantly from that of the females. The mean ages of the discharged patients were 64.4 ± 16.2 years (male) and 69.2 ± 14.3 years (female). There were 199 (23.4%) deaths, of which 126 (63.3%) decedents were male and 73 (36.7%) were female. The mean age of the patients who died was 69.2 ± 14.3.

The type of organ failure at ICU admission were as follows: lung = 44.5%, CV = 36.1%, CNS = 32.7%, liver = 11.2% and kidney = 7.6%. Anonymous data are available in the repository as *DATASET SOFA*. DANS. https://doi.org/10.17026/dans-zgz-7keg.

**Networks.** Based on the bootstrap analysis, strength and direction values were robust if they exceeded the threshold value of 0.47 for strength and we assigned the conventionally adopted value of 0.5 for direction. The first network (6 nodes for 8 days) had 54 nodes and 74 direct arcs, with average Markov blanket size of 3.33. The $2^{nd}$ network included 45 nodes and 60 direct arcs, with a Markov blanket size of 3.16. The $3^{rd}$ network showed 18 nodes and 22 direct arcs, a Markov blanket size of 3.00 and a cutoff value of 0.64 when "C" was included, whereas without the "C" node, the number of nodes was reduced to 15 and the number of arcs to 19, with a Markov blanket size of 3.20. The network analysis produced the following findings: after bootstrapping, the structure of the $1^{st}$ network deviated from the original one and the strengths of 12 arcs and 15 directions were lower than the cutoff value of 0.49. In the $2^{nd}$ network, 2 of 60 arcs deviated from the original structure after bootstrapping, and 9 arcs and 14 directions were below the strength cutoff of 0.48.

In the $3^{rd}$ network including C, 3 of 22 arcs differed from the original structure after bootstrapping, and 6 arcs and 8 directions were below the strength cutoff of 0.48. When C was excluded, the structure of the $3^{rd}$ network differed from the original in four arcs after bootstrapping (Fig 1). The final network had 15 nodes, 16 arcs, and an average Markov blanket size of 2.93.

Among the four tested networks, the fourth one, which included 5 variables measured at $t0$, $t2$ and t7, had a strength of more than 0.67 for all arcs, and the minimum number of time points still retained relevant clinical information. In Table 1, we included parameters (namely,

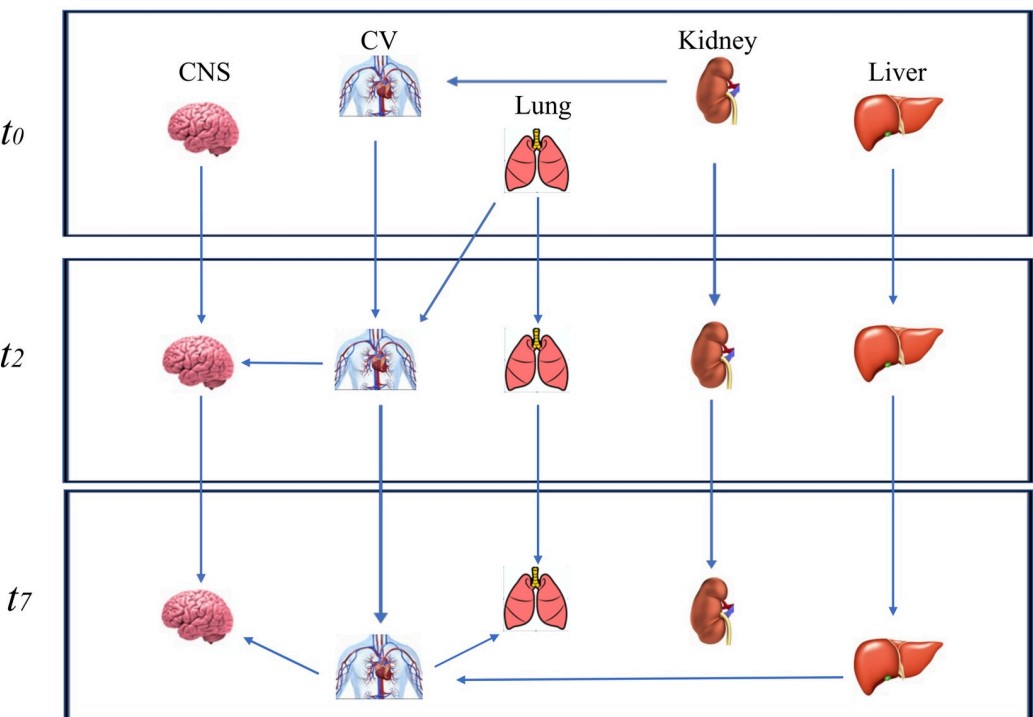

**Fig 1. Dynamic Bayesian network excluding the coagulation node.** The Dynamic Bayesian network without the coagulation node (C) at three time points ($t0$ = the day of ICU admission; $t2$ = 2nd day and $t7$ = 7th day) after bootstrapping.

**Table 1. Arcs strength and direction of the averaged dynamic Bayesian network obtained by 500 bootstrap on five organs at the intensive care unit admission, at 2nd and 7th day.**

| From | To | strength | direction |
|---|---|---|---|
| $Kidney_{t0}$ | $Kidney_{t2}$ | 1.00 | 0.79 |
| $Kidney_{t0}$ | $Lung_{t2}$ | 1.00 | 1.00 |
| $Liver_{t0}$ | $Liver_{t2}$ | 0.98 | 0.62 |
| $Lung_{t0}$ | $Lung_{t2}$ | 1.00 | 1.00 |
| $Lung_{t0}$ | $Cardiovascular_{t2}$ | 1.00 | 1.00 |
| $Cardiovascular_{t0}$ | $Cardiovascular_{t2}$ | 1.00 | 1.00 |
| $Central\ Nervous\ System_{t0}$ | $Central\ Nervous\ System_{t2}$ | 1.00 | 1.00 |
| $Kidney_{t2}$ | $Kidney_{t7}$ | 1.00 | 0.94 |
| $Liver_{t2}$ | $Liver_{t7}$ | 1.00 | 0.80 |
| $Liver_{t2}$ | $Cardiovascular_{t7}$ | 1.00 | 1.00 |
| $Lung_{t2}$ | $Lung_{t7}$ | 1.00 | 1.00 |
| $Cardiovascular_{t2}$ | $Central\ Nervous\ System_{t2}$ | 0.67 | 0.91 |
| $Cardiovascular_{t2}$ | $Cardiovascular_{t7}$ | 1.00 | 0.99 |
| $Central\ Nervous\ System_{t2}$ | $Central\ Nervous\ System_{t7}$ | 1.00 | 0.98 |
| $Cardiovascular_{t7}$ | $Lung_{t7}$ | 0.94 | 0.79 |
| $Cardiovascular_{t7}$ | $Central\ Nervous\ System_{t7}$ | 0.96 | 0.85 |

The day of the intensive care unit admission: $t_0$; the 2nd day of intensive care unit stay: $t_2$; the 7th day of intensive care unit stay: $t_7$.

**Table 2. The sensitivity and specificity of the final network prediction tested with the *k*-fold cross-validation analysis.**

| Variable | sensitivity | specificity |
|---|---|---|
| Central Nervous System $t_2$ | 0.90 | 0.88 |
| Kidney $t_2$ | 0.76 | 0.98 |
| Lung $t_2$ | 0.70 | 0.69 |
| Cardiovascular $t_2$ | 0.79 | 0.83 |
| Liver $t_2$ | 0.32 | 0.96 |
| Central Nervous System $t_7$ | 0.80 | 0.89 |
| Kidney $t_7$ | 0.72 | 0.98 |
| Lung $t_7$ | 0.21 | 0.97 |
| Cardiovascular $t_7$ | 0.21 | 0.99 |
| Liver $t_7$ | 0.70 | 0.99 |

The 2[nd] day of intensive care unit stay: $t_2$; the 7[th] day of intensive care unit stay: $t_7$.

the strength and direction) to estimate the accuracy of the DBN regarding five organs evaluated at t0 and on the 2[nd] and 7[th] days.

The reliability of this network prediction was supported by the cross-validation results which indicated high sensitivity and specificity for almost all nodes (Table 2).

**Conditional probabilities.** We fitted the conditional probability distributions for the arcs of the last network with five variables (excluding platelets), and three time periods (admission, 2[nd] day and 7[th] day). The resulting probability matrices are reported in S1–S3 Figs.

Kidney and liver failure at the time of admission had a 72% and 51% probability, respectively, of persisting to the 2[nd] day. The probabilities of these organ failures persisting from the 2[nd] day to the 7[th] day were higher, at 86% and 69%, respectively. CV failure had a 70% probability of persisting to the 2[nd] day when associated with lung failure on ICU admission, whereas the probability decreased to 66% in the absence of this association. Similarly, lung failure on admission had a 69% probability of persisting to the 2nd day when associated with kidney failure, whereas this probability was only 49% in the absence of this association. CNS failure on admission had a 71% probability of persisting to the 2nd day, but when associated with CV on the 2nd day, its probability increased to 75%. The probability of CNS failure persisting from the 2[nd] to the 7[th] day was 60%, but it increased to 86% when CV failure was present on the 7[th] day. The probability that CV failure on the 2[nd] day would persist to the 7[th] day was 72% in association with liver failure but 62% in the absence of this association. Finally, lung failure on the 2[nd] day showed a 58% probability of persisting to the 7[th] day, but the probability increased to 67% when CV failure was also present. In Fig 2, we report the types of organ failure and their associations with increased probabilities of progression during the observation time of our study.

**Network reliability.** Our results demonstrated the feasibility of achieving a sufficiently reliable DBN model to predict the association of organ failures and their evolution regardless of a predetermined final outcome. Our previous study focused on predicting sequences of organ failure but used a node "discharge" in the DBN [6]. The current study overcomes the limitation of using a predefined outcome and allows the network structure to learn from data. In addition, we used a learning algorithm strategy to estimate the conditional probabilities.

Defining an optimal network structure that learns its parameters is a complex computational problem. The number of structures to be tested can be large, and providing a good estimate of probabilities requires a large volume of data to be processed. In order to define the

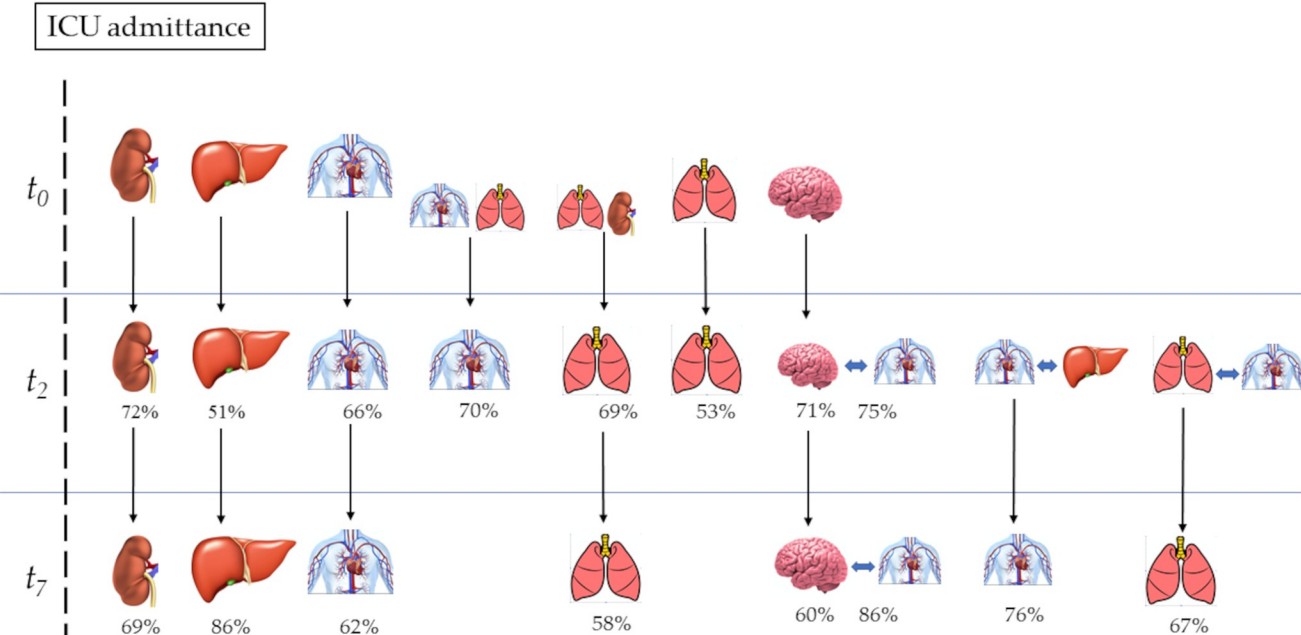

**Fig 2. Percentage probabilities of the organ failure progression.** The day of the intensive care unit admission: $t_0$; the 2nd day after the intensive care unit admission: $t_2$; the 7th day after the intensive care unit admission: $t_7$. Percentage numbers: conditioned probability to progress at $2^{nd}$ and 7 day$^{th}$.

network structure that best describes data and relationships among variables, two different approaches can be used: a "global" approach [21], which employs search and score algorithms for data description, and a "local" approach [22, 23], which utilizes conditional independent tests to evaluate relationships. This latter approach derives the best structure representing relationships. The search and score methods proceed by creating several network structures suitable to describe data and assign scores to them with a specific scoring function. After comparing scores, a search algorithm identifies the most representative network structure. These steps are needed to limit the number of networks to be evaluated, reducing the computational load.

In our study, we applied the HC heuristic research method, a "local" approach that is applied when the graph is unknown [18], to improve data fitting despite the large number of organs (6) and the long examination period (7 days). To obtain the best network fit and increase its accuracy, we reduced the number of timepoints instead of organs, leaving information on organ failure events unchanged.

To reduce the network complexity due to the number of organ, we removed only the "coagulation" metric, as platelet counts during sepsis should ordinary be increased by supportive actions focused on fully treating the spread of infection than by a specific intervention [24].

**Network accuracy.** We found several organ failure associations with an increased probability of occurring and progressing over time. Our model reported only conditional probabilities resulting from arcs between organs. A network with three time points and five organs proved to be sufficiently accurate for our intended clinical purposes. At the time of ICU admission, a real connection between organs was unlikely to occur, and there is a low probability of evidence connecting the failure of these organs. In contrast, the persistence of organ failure during the ICU stay was more probable when multiple failing organs were associated.

Information derived from our network model provides incentives for clinical reasoning. Any clinical reasoning should be based on an understanding of the relationships among the

elements that are relevant to the individual clinical case; thus, the characteristics of the elements forming the basis of the reasoning assessment become crucial. When BNs associate probabilities with a single outcome (life or death) or composite outcomes, they furnish information similar to what is derived from severity scores, but the Bayesian perspective can supply other elements, in terms of organ failure, that allow enrichment of clinical reasoning. In fact, it is likely that the associations and sequences of organ failure in a given population result from treatments and clinical approaches adopted by clinicians. Avoiding predefined outcomes leads to other perspectives, such as adopting therapeutic strategies to change the scenario of organ damage.

Our study has several limitations. We restricted our node to the presence of organ failure, neglecting its severity. The need to attribute a weight to the organ associations rather than to the severity of failure has limited the Bayesian model we adopted. Adopting a graded intensity scale for organ damage could increase the available clinical information in the near future while not interfering with the ability to detect probabilities of organ relationships. Another factor that could interfere with the real evaluation of organ failure is external support given to organ function. This issue is also present for all the predictive scores [25] and leads to the consideration of external organ support as an integral part of organ function evaluation. Finally, given the size of our dataset, we built a robust network structure using only three time slices and five organs. More specifics regarding the time course of organ failure may be useful with a larger number of patients.

## Conclusions

In this study, we applied an innovative approach for testing the reliability and accuracy of a DBN model aimed to avoid imposing predefined outcomes. We realized a network model that allowed us to predict with satisfactory accuracy several organ failure associations and their evolution in critically ill patients. As the organ failure sequences likely resulted from the clinical choices adopted, our method has the potential advantage of detecting the effects of treatments or therapeutic strategies on organ function and comparing these effects on populations treated differently. Further analysis is needed to test the accuracy of a network model able to assess the severity of organ dysfunction and determine whether it could add useful knowledge in clinical settings.

## Supporting information

**S1 Fig. Percentage conditioned probability of failing organs and organs associations at the intensive care unit admittance.** Horizontal axis: percentage probability. Vertical axis: present 1. Absent 0. Intensive care unit admittance: t0. Organ on the rectangles border: associated organ.
(TIF)

**S2 Fig. Percentage conditioned probability of failing organs and organs associations at the 2nd day of the intensive care stay.** Horizontal axis: percentage probability. Vertical axis: present 1. Absent 0. Intensive care unit admittance: t0. Organ on the rectangles border: associated organ.
(TIF)

**S3 Fig. Percentage conditioned probability of failing organs and organs associations at the 7th day of the intensive care stay.** Horizontal axis: percentage probability. Vertical axis: present 1. Absent 0. Intensive care unit admittance: t0. Organ on the rectangles border: associated

organ.
(TIF)

## Acknowledgments

We acknowledge dr. Mario Tavola (chief of the Intensive Care Unit of the A. Manzoni Hospital of Lecco—Italy) and dr. Romano Tetamo (chief of the Anesthesia and Intensive Care Medicine of the Di Cristina-Benfratelli Civic Hospital of Palermo—Italy) for having made available data from patients admitted in their Units.

## Author Contributions

**Conceptualization:** Roberto Alberto De Blasi.

**Data curation:** Roberto Alberto De Blasi, Stefano Finazzi.

**Formal analysis:** Giuseppe Campagna.

**Funding acquisition:** Roberto Alberto De Blasi.

**Investigation:** Roberto Alberto De Blasi.

**Methodology:** Roberto Alberto De Blasi, Giuseppe Campagna, Stefano Finazzi.

**Validation:** Stefano Finazzi.

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
