## [Decision Letter · Decision Letter 0]

23 Dec 2020

PONE-D-20-33480

A dynamic Bayesian network model for predicting organ failure associations without predefining outcomes

PLOS ONE

Dear Dr. De Blasi,

Thank you for submitting your manuscript to PLOS ONE. After careful consideration, we feel that it has merit but does not fully meet PLOS ONE’s publication criteria as it currently stands. Therefore, we invite you to submit a revised version of the manuscript that addresses the points raised during the review process.

We look forward to receiving your revised manuscript.

Kind regards,

Moshe Zukerman

Academic Editor

PLOS ONE

Journal Requirements:

2. Please provide additional details regarding participant consent. In the ethics statement in the Methods and online submission information, please ensure that you have specified whether consent was informed. If the need for additional consent was waived by the ethics committee, please include this information.

3. In the ethics statement in the manuscript and in the online submission form, please provide additional information about the patient records/samples used in your retrospective study, including: a) whether all data were fully anonymized before you accessed them; b) the date range (month and year) during which patients' medical records/samples were accessed.

4. In your Methods section, please provide additional information about the medical data/samples collected and the demographic details of the human subjects. Please ensure you have provided sufficient details to replicated the analyses such as a table of relevant demographic details.

6. Please upload a new copy of Figure 1 as the detail is not clear. Please follow the link for more information: https://blogs.plos.org/plos/2019/06/looking-good-tips-for-creating-your-plos-figures-graphics/

Reviewers' comments:

Reviewer's Responses to Questions

**Comments to the Author**

1. Is the manuscript technically sound, and do the data support the conclusions?

Reviewer #1: Yes

Reviewer #2: Partly

2. Has the statistical analysis been performed appropriately and rigorously? 

Reviewer #1: Yes

Reviewer #2: I Don't Know

3. Have the authors made all data underlying the findings in their manuscript fully available?

Reviewer #1: Yes

Reviewer #2: No

4. Is the manuscript presented in an intelligible fashion and written in standard English?

Reviewer #1: Yes

Reviewer #2: No

5. Review Comments to the Author

Reviewer #1: This paper represents a very interesting study with informative results. The overall structure of the paper is acceptable; however, it can be improved based on deploying the below ideas.

1. Using model stacking to improve accuracy as the model stacking maintain the interpretability of the DBN.

2. How the DBN network is initialized? I could not find any description or sentence that explains the network initialization before training?

3. Please use more creative approaches to illustrate your results. Since the paper claims at beginning, that the interpretability is one of their goals, but no data visualization approach is used for presenting the results.

4. Please provide a link to your code of the model in a public repository.

Reviewer #2: General comments:

Interesting and impactful application of DBN for a key problem as demonstrated by the discussions in the section beginning line 203. However, there are several key areas that are lacking in clarity, and major revision at minimum is needed to ensure that the research is significant, original and well justified.

The paper could benefit significantly from detailed review and proof-reading as sentence construction and grammar is at times quite poor. There are many instances (some examples are provided below) where:

(i) clarification is needed, and/or

(ii) citations / references are needed to justify a statement, and/or

(iii) justification of a design decision is needed.

There is insufficient detail about the implementation of the DBN and Figure 1 is illegible to enable scientific reproducibility or indeed assessment of the proposed work.

This model seems to be about *short term* multiple organ failure that occurs over the course of a days (6 days) – this needs to be clarified.

A key question for the developed models, however, is model validation. Validation appears to be missing even though line 191 alludes to network C “satisfy(ing) the accuracy requirements”. Without providing validation results, the reported findings are not necessarily reliable.

47: what do you mean by “…systems in which the relationship between variables is not completely known”? Are you studying and evaluating different graph structures like that done in BN learning (Daly, Shen, & Aitken, 2011)?

53: what do you mean here? Are you talking about using a model to determine if a therapeutic action contributed to a given outcome? Please clarify.

61: how did you select variables to include? What are the “values” to be included, are you referring to node states?

63: Clarify – was the graph entirely built using expert elicitation?

The description of the procedure you followed to build the network needs some justification and/or citations. Some examples of typical BN construction methodologies are discussed by (Johnson et al., 2010; Korb and Nicholson, 2010; Pollino, Woodberry, Nicholson, Korb, & Hart, 2007).

72: also needs references for these statements

The paper needs some review of existing studies and/or models of single and/or multi-organ failures – how are these failures currently modelled? What is the gap? Hence, what is the significance and contribution of the proposed approach?

90: why did you exclude young patients <18 years old and short hospital stays? What is the consequence of focusing on 2 or more organ failures?

96: what is i indexing, the patient?

100: “…considered to be failing if it had a score of >=2” – needs a citation

105: kNN for data imputation – however, the Expectation Maximisation (EM) algorithm typically used in BN learning is able to impute missing data using the entire BN and other data points. Why is it necessary to use kNN?

119: Markov assumption, not Markov blanket

125: Not clear how you used bootstrapping – please describe briefly the procedure you used to learn the DBN and how you used the data.

Figure 1 is illegible

129: it is unclear how you have gone about building these different networks, how you used hill climbing, and how your approach is justified. This section needs much better clarity.

137: It is well known that a blacklist to prevent illogical arcs and a whitelist to force arcs is commonly needed and is supported by bnlearn and the papers around it – these should be cited. Also, for reproducibility, the specific blacklist and whitelists should be provided and justified.

REFERENCES

Daly, R., Shen, Q., & Aitken, S. (2011). Learning Bayesian networks: approaches and issues. The Knowledge Engineering Review, 26(02), pp. 99-157. doi:doi:10.1017/S0269888910000251 Retrieved from http://dx.doi.org/10.1017/S0269888910000251

Johnson, S., Mengersen, K., Waal, A. d., Marnewick, K., Cilliers, D., Houser, A. M., & Boast, L. (2010). Modelling cheetah relocation success in southern Africa using an Iterative Bayesian Network Development Cycle. Ecological Modelling, 221, pp. 641-651.

Korb, K. B., & Nicholson, A. E. (2010). Bayesian Artificial Intelligence, Second Edition: CRC Press, Inc.

Pollino, C. A., Woodberry, O., Nicholson, A., Korb, K., & Hart, B. T. (2007). Parameterisation and evaluation of a Bayesian network for use in an ecological risk assessment. Environmental Modelling and Software, 22(8), pp. 1140-1152.

6. PLOS authors have the option to publish the peer review history of their article (what does this mean?). If published, this will include your full peer review and any attached files.

Reviewer #1: **Yes: **Omid Bazgir

Reviewer #2: **Yes: **Paul Pao-Yen Wu

---

## [Author Response · Author response to Decision Letter 0]

22 Feb 2021

Reviewer #1: This paper represents a very interesting study with informative results. The overall structure of the paper is acceptable; however, it can be improved based on deploying the below ideas.

1. Using model stacking to improve accuracy as the model stacking maintain the interpretability of the DBN.

We agree with the reviewer that model stacking should improve the accuracy of model prediction by efficiently aggregating the results of several models. Nevertheless, as our primary aim was proving that the BN structure for failing organs probabilistic interaction and evolution was robust and led to a clinically sensible interpretation, we believe that a simpler boostrap validation of the model structure can be appropriate. In this manuscript revision, we have extended Materials and methods and better explained how we have generated the network structure and the boostrap validation in - Dynamic Bayesian Network formulation (page 5, lines 135-145). 

2. How the DBN network is initialized? I could not find any description or sentence that explains the network initialization before training?

The network structure was learned from data using the hill-climbing algorithm provided by the R package bnlearn, starting from an empty network (a network with no arcs). The score used in the optimization process was the Bayesian information criterion. We have provided these details about network construction in the section Materials and methods - Dynamic Bayesian Network formulation (lines 128-135). We have also added in methods how we estimated the network probability matrices on the arcs. This is equivalent to using a maximum a posteriori estimation from a uniform prior distribution of the parameters.

3. Please use more creative approaches to illustrate your results. Since the paper claims at beginning, that the interpretability is one of their goals, but no data visualization approach is used for presenting the results. 

We apologize for the scarcity of our results visualization. In this revised version we have modified the Bayesian network in figure 1 to make it clearer (line 182) and added another figure of the most significant conditioned probabilities for the organ failure onset, their associations and progressions (figure 2) (line 226). For further information, we have also added, as supporting figures, extended documentation of the percentage probabilities for the organ failure and their associations at the three time points (line 210).

4. Please provide a link to your code of the model in a public repository. 

We have sent our dataset to the public repository DANS-EASY Electronic Archiving System DANS.https://doi.org/10.17026/dans-zgz-7keg (Page 7, lines 163).

Reviewer #2: General comments:

Interesting and impactful application of DBN for a key problem as demonstrated by the discussions in the section beginning line 203. However, there are several key areas that are lacking in clarity, and major revision at minimum is needed to ensure that the research is significant, original and well justified.

The paper could benefit significantly from detailed review and proof-reading as sentence construction and grammar is at times quite poor. There are many instances (some examples are provided below) where:

(i) clarification is needed, and/or

(ii) citations / references are needed to justify a statement, and/or

(iii) justification of a design decision is needed.

There is insufficient detail about the implementation of the DBN and Figure 1 is illegible to enable scientific reproducibility or indeed assessment of the proposed work.

We have improved the description of procedures adopted to implement the DBN in methods with references to the R-package we used (Page 5, lines 128).

This model seems to be about *short term* multiple organ failure that occurs over the course of a days (6 days) – this needs to be clarified. 

We have accepted the reviewer’s suggestions: “short term” has been added in the introduction (line 80) and the reason why our study extended over a period of 7 days has been reported in Methods (line 105).

A key question for the developed models, however, is model validation. Validation appears to be missing even though line 191 alludes to network C “satisfy(ing) the accuracy requirements”. Without providing validation results, the reported findings are not necessarily reliable.

We are aware that model validation has a key role for findings reliability. In this revision, we have applied a k-fold cross-validation analysis to the final network. We computed sensitivities and specificities of the predictions of child nodes from parent nodes. We have added several details in Methods for the cross-validation analysis and a table (table 2) (line 201) showing predictive sensitivity and specificity. 

47: what do you mean by “…systems in which the relationship between variables is not completely known”? Are you studying and evaluating different graph structures like that done in BN learning (Daly, Shen, & Aitken, 2011)? 

We have rephrased the first sentence of introduction clarifying the BNs characteristics we considered for our study including the suggested reference [2].

53: what do you mean here? Are you talking about using a model to determine if a therapeutic action contributed to a given outcome? Please clarify.

We agree with the reviewer that sentences (line 53-61) needed to be clarified. We have now specified that we considered probabilities of failing organs as reflecting the effects of treatments (lines 52-56). In the Conclusions we have resumed this concept and extended its possible clinical applications (lines 294-296).

61: how did you select variables to include? What are the “values” to be included, are you referring to node states? 

63: Clarify – was the graph entirely built using expert elicitation? 

The paper needs some review of existing studies and/or models of single and/or multi-organ failures – how are these failures currently modelled? What is the gap? Hence, what is the significance and contribution of the proposed approach? 

Based on the reviewer observations, we have realized that paragraph from line 61 was confusing and poorly comprehensible. In this version we have completely revised it, anticipating the aim of our work and highlighting differences with existing studies and contribution of our approach (line 60-83). We have also added references, reduced sentences, corrected unclear terms and attempted to make the text clearer.

The description of the procedure you followed to build the network needs some justification and/or citations. Some examples of typical BN construction methodologies are discussed by (Johnson et al., 2010; Korb and Nicholson, 2010; Pollino, Woodberry, Nicholson, Korb, & Hart, 2007). 

72: also needs references for these statements 

We have provided more details about the procedures we followed to learn the network structure and fit its conditional probabilities including references in Materials and methods - Dynamic Bayesian Network formulation. We added as supporting information the resulting probability matrices (S1-S3). 

90: why did you exclude young patients <18 years old and short hospital stays? What is the consequence of focusing on 2 or more organ failures? 

We regret for not having reported that the study was realized in adult ICUs and no patients under 18 years old had criteria to be included in the study. In this revision we have clarified this (line 89) and deleted “under 18 years old” from the exclusion criteria. We have also explained why we focused on 2 or more organ failures (line 92).

100: “…considered to be failing if it had a score of >=2” – needs a citation

We have included a reference to the sentence (line 104).

105: kNN for data imputation – however, the Expectation Maximisation (EM) algorithm typically used in BN learning is able to impute missing data using the entire BN and other data points. Why is it necessary to use kNN? 

We agree with the reviewer in considering the kNN not necessary. Therefore, we have deleted the kNN from the manuscript.

119: Markov assumption, not Markov blanket

We have changed “blanket” in “assumption” as suggested (line 123). 

125: Not clear how you used bootstrapping – please describe briefly the procedure you used to learn the DBN and how you used the data.

In this manuscript revision, we have clarified how we used bootstrapping by adding the description of procedures we adopted to learn the DBNs (page 5, line 128) and a reference [19] (Nagarajan R, Scutari M, Lèbre S. Bayesian Networks in R. New York: Springer-Verlag; 2013). 

Figure 1 is illegible:

We have changed figure 1 attempting to make it clearer for readers.

129: it is unclear how you have gone about building these different networks, how you used hill climbing, and how your approach is justified. This section needs much better clarity. 

We have added a detailed description on the approach we used for the network construction and hill climbing in Methods (page 5, line 129-138).

137: It is well known that a blacklist to prevent illogical arcs and a whitelist to force arcs is commonly needed and is supported by bnlearn and the papers around it – these should be cited. Also, for reproducibility, the specific blacklist and whitelists should be provided and justified. 

In this revision, we have disclosed in the introduction (line 80) the use of the blacklist we adopted in learning the DBN structure according to the aims of our work and provided a description of the blacklist characteristics in Methods-Dynamic Bayesian Network formulation (line 131).

---

## [Decision Letter · Decision Letter 1]

16 Mar 2021

PONE-D-20-33480R1

A dynamic Bayesian network model for predicting organ failure associations without predefining outcomes

PLOS ONE

Dear Dr. De Blasi,

Thank you for submitting your manuscript to PLOS ONE. After careful consideration, we feel that it has merit but does not fully meet PLOS ONE’s publication criteria as it currently stands. Therefore, we invite you to submit a revised version of the manuscript that addresses the points raised during the review process.

We look forward to receiving your revised manuscript.

Kind regards,

Moshe Zukerman

Academic Editor

PLOS ONE

Journal Requirements:

Reviewers' comments:

Reviewer's Responses to Questions

**Comments to the Author**

1. If the authors have adequately addressed your comments raised in a previous round of review and you feel that this manuscript is now acceptable for publication, you may indicate that here to bypass the “Comments to the Author” section, enter your conflict of interest statement in the “Confidential to Editor” section, and submit your "Accept" recommendation.

Reviewer #1: All comments have been addressed

Reviewer #2: (No Response)

2. Is the manuscript technically sound, and do the data support the conclusions?

Reviewer #1: Partly

Reviewer #2: Yes

3. Has the statistical analysis been performed appropriately and rigorously? 

Reviewer #1: Yes

Reviewer #2: Yes

4. Have the authors made all data underlying the findings in their manuscript fully available?

Reviewer #1: No

Reviewer #2: Yes

5. Is the manuscript presented in an intelligible fashion and written in standard English?

Reviewer #1: Yes

Reviewer #2: Yes

6. Review Comments to the Author

Reviewer #1: The authors took care of the my previous comments pretty well and all my concerns in the previous round of review were addressed in the current revision.

Reviewer #2: The authors have done an admirable job addressing the comments and the paper is offers more comprehensive detail and clarity on the approach. I appreciate the inclusion of the conditional probability tables in the supporting information too, that is important for reproducibility of the research. I still think that the paper would benefit from some final, professional, proof-reading. I have attached below some minor clarifications:

Line 69: I understand that imposing constraints may limit the ability of the model to reflect the data; however, the data itself may not truly represent the complex system that is MODS.

I would suggest: “On the other hand, imposing an ordering and constraints on the probabilistic relationships between organs and processes can limit the ability of the model to reflect the data.”

Line 75: “…dynamics of failing organs by hierarcizing organ interactions and forcing a discrete outcome… ”

Line 81: “…structure of a short-term DBN from data with Markovian constraints enforced through a blacklist.”

Line 103: did you mean “A failure event for an organ was defined as a SOFA score of greater than or equal to two”.

Line 133: “and enforce Markovian properties…”

Line 136 paragraph – very good clarification!

Line 151 – what is node C, is it the platelet number (if so, say that here)

Line 187 – need to check grammar here!

Line 260 – loops violating the Directed Acyclic Graph (DAG) assumption of BNs. Markov assumption is something else.

Sometimes there is shorthand (e.g. L instead of Lung) – please be consistent

7. PLOS authors have the option to publish the peer review history of their article (what does this mean?). If published, this will include your full peer review and any attached files.

Reviewer #1: **Yes: **Omid Bazgir

Reviewer #2: No

---

## [Author Response · Author response to Decision Letter 1]

1 Apr 2021

To: Moshe Zukerman

Academic Editor

PLOS ONE

Dear Editor,

we have revised the manuscript responding to each point raised by the reviewer #2. The manuscript has undergone a further English editing by the SpringerNature Author Service company on March 23, 2021 (verification code certification: 906E-C0AA-00AD-B5B9-4803). In the 'Revised Manuscript with Track Changes' we have corrected some grammar errors and highlighted the changes of the English form (yellow) and the changes suggested by the reviewer (green). We have not modified references or figures.

Best regards.

Roberto A. De Blasi

Review Comments to the Author

Reviewer #2: The authors have done an admirable job addressing the comments and the paper is offers more comprehensive detail and clarity on the approach. I appreciate the inclusion of the conditional probability tables in the supporting information too, that is important for reproducibility of the research. I still think that the paper would benefit from some final, professional, proof-reading. I have attached below some minor clarifications:

Line 69: I understand that imposing constraints may limit the ability of the model to reflect the data; however, the data itself may not truly represent the complex system that is MODS.

I would suggest: “On the other hand, imposing an ordering and constraints on the probabilistic relationships between organs and processes can limit the ability of the model to reflect the data.”

We have revised line 69 as suggested by the reviewer (line 70).

Line 75: “…dynamics of failing organs by hierarcizing organ interactions and forcing a discrete outcome.. ”

We have accepted the reviewer’s suggestion.

Line 81: “…structure of a short-term DBN from data with Markovian constraints enforced through a blacklist.”

We have revised line 79 taking into account the reviewer’s suggestion.

Line 103: did you mean “A failure event for an organ was defined as a SOFA score of greater than or equal to two”.

We have revised line 103 as suggested by the reviewer because it’s more comprehensible

Line 133: “and enforce Markovian properties…”

We have corrected the sentence at line 133 as suggested (line 131).

Line 151 – what is node C, is it the platelet number (if so, say that here)

We have clarifies that node C refers to the platelet count (line 149).

Line 187 – need to check grammar here!

We have completely revised the sentence (line 185-187).

Line 260 – loops violating the Directed Acyclic Graph (DAG) assumption of BNs. Markov assumption is something else.

As the sentence was not conceptually correct and did not add substantial knowledge at the discussion, we have deleted the whole sentence at lines 259-261. 

Sometimes there is shorthand (e.g. L instead of Lung) – please be consistent

As the “L” abbreviation could be confounding, we have deleted this shorthand and inserted the full name “liver”

---

## [Decision Letter · Decision Letter 2]

14 Apr 2021

A dynamic Bayesian network model for predicting organ failure associations without predefining outcomes

PONE-D-20-33480R2

Dear Dr. De Blasi,

We’re pleased to inform you that your manuscript has been judged scientifically suitable for publication and will be formally accepted for publication once it meets all outstanding technical requirements.

Kind regards,

Moshe Zukerman

Academic Editor

PLOS ONE

Additional Editor Comments (optional):

Reviewers' comments:

Reviewer's Responses to Questions

**Comments to the Author**

1. If the authors have adequately addressed your comments raised in a previous round of review and you feel that this manuscript is now acceptable for publication, you may indicate that here to bypass the “Comments to the Author” section, enter your conflict of interest statement in the “Confidential to Editor” section, and submit your "Accept" recommendation.

Reviewer #2: All comments have been addressed

2. Is the manuscript technically sound, and do the data support the conclusions?

Reviewer #2: Yes

3. Has the statistical analysis been performed appropriately and rigorously? 

Reviewer #2: Yes

4. Have the authors made all data underlying the findings in their manuscript fully available?

Reviewer #2: Yes

5. Is the manuscript presented in an intelligible fashion and written in standard English?

Reviewer #2: Yes

6. Review Comments to the Author

Reviewer #2: The comments have been adequately addressed from the two rounds of review and revisions, thank you!

7. PLOS authors have the option to publish the peer review history of their article (what does this mean?). If published, this will include your full peer review and any attached files.

Reviewer #2: **Yes: **Paul Pao-Yen Wu

---

## [Editor Report · Acceptance letter]

16 Apr 2021

PONE-D-20-33480R2 

A dynamic Bayesian network model for predicting organ failure associations without predefining outcomes 

Dear Dr. De Blasi:

I'm pleased to inform you that your manuscript has been deemed suitable for publication in PLOS ONE. Congratulations! Your manuscript is now with our production department. 

Kind regards, 

on behalf of

Dr. Moshe Zukerman 

Academic Editor

PLOS ONE